# What Is the Best Practice for CNNs Applied to Visual Instance Retrieval?

**Jiedong Hao, Jing Dong, Wei Wang, Tieniu Tan**
Center for Research on Intelligent Perception and Computing
Institute of Automation, Chinese Academy of Sciences

## Abstract

Previous work has shown that feature maps of deep convolutional neural networks (CNNs) can be interpreted as feature representation of a particular image region. Features aggregated from these feature maps have been exploited for image retrieval tasks and achieved state-of-the-art performances in recent years. The key to the success of such methods is the feature representation. However, the different factors that impact the effectiveness of features are still not explored thoroughly. There are much less discussion about the best combination of them.

The main contribution of our paper is the thorough evaluations of the various factors that affect the discriminative ability of the features extracted from CNNs. Based on the evaluation results, we also identify the best choices for different factors and propose a new multi-scale image feature representation method to encode the image effectively. Finally, we show that the proposed method generalises well and outperforms the state-of-the-art methods on four typical datasets used for visual instance retrieval.

## 1 Introduction

Image retrieval is an important problem both for academic research and for industrial applications. Although it has been studied for many years (Sivic & Zisserman, 2003; Philbin et al., 2007; Tolias et al., 2015), it is still a challenging task. Generally, image retrieval is divided into two groups. The first one is the category-level image retrieval (Sharma & Schiele, 2015), in which an image in the dataset is deemed to be similar to the query image if they share the same class or they are similar in shape and local structures. The other group is the instance-level image retrieval (Tolias et al., 2015), in which an image is considered to match the query if they contain the same object or the same scene. The instance-level image retrieval is harder in that the retrieval method need to encode the local and detailed information in order to tell two images apart, *e.g.*, the algorithm should be able to detect the differences between the Eiffel Tower and other steel towers although they have similar shapes. In this paper, we focus on the instance-level image retrieval.

Traditionally, visual instance retrieval is mainly addressed by the BoF (bag of features) based methods using the local feature descriptors such as SIFT (Lowe, 2004). In order to boost the retrieval performances, post-processing techniques such as query expansion (Chum et al., 2007) and spatial verification (Philbin et al., 2007) are also employed.

With the decisive victory (Krizhevsky et al., 2012) over traditional models in the ImageNet (Russakovsky et al., 2015) image classification challenge, convolutional neural networks (Lecun et al., 1998) continue to achieve remarkable success in diverse fields such as object detection (Liu et al., 2015; Shaoqing Ren, 2015), semantic segmentation (Dai et al., 2016) and even image style transfer (Gatys et al., 2016). Networks trained on the Imagenet classification task can generalize quite well to other tasks, which are either used off-the-shelf (Razavian et al., 2014a) or fine-tuned on the task-specific datasets (Azizpour et al., 2014; Long et al., 2015). Inspired by all these, researchers in the field of image retrieval also shift their interest to the CNNs. Their experiments have shown promising and surprising results (Babenko et al., 2014; Razavian et al., 2014c; Tolias et al., 2015), which are on par with or surpass the performances of conventional methods like BoF and VLAD (vector of locally aggregated descriptors) (Jégou et al., 2010; Arandjelović & Zisserman, 2013) .

Despite all these previous advances (Babenko et al., 2014; Babenko & Lempitsky, 2015; Tolias et al., 2015) on using CNNs for image feature representation, the underlying factors that contribute to the success of off-the-shelf CNNs on the image retrieval tasks are still largely unclear and unexplored, *e.g., which layer is the best choice for instance retrieval, the convolutional layer or the fully-connected layer? What is the best way to represent the multi-scale information of an image?* Clarifying these questions will help us advance a further step towards building a more robust and accurate retrieval system. Also in situations where a large numbers of training samples are not available, instance retrieval using unsupervised method is still preferable and may be the only option.

In this paper, we aim to answer these questions and make three novel contributions. Unlike previous papers, we explicitly choose five factors to study the image representations based on CNNs and conduct extensive experiments to evaluate their impacts on the retrieval performances. We also give detailed analysis on these factors and give our recommendations for combining them. During experiments, we borrow wisdoms from literatures and evaluate their usefulness, but find that they are not as effective as some of the simpler design choices. Second, by combining the insights obtained during the individual experiments, we are able to propose a new multi-scale image representation, which is compact yet effective. Finally, we evaluate our method on four challenging datasets, *i.e.*, Oxford5k, Paris6k, Oxford105k and UKB. Experimental results show that our method is generally applicable and outperforms all previous methods on compact image representations by a large margin.

## 2 RELATED WORK

**Multi-scale image representation**. Lazebnik et al. (2006) propose the spatial pyramid matching approach to encode the spatial information using BoF based methods. They represent an image using a pyramid of several levels or scales. Features from different scales are combined to form the image representation in such a way that coarser levels get less weight while finer levels get more weight. Their argument is that matches found in coarser levels may involve increasingly dissimilar image features. In our paper, we also explore the multi-scale paradigm in the same spirit using the convolutional feature maps as the local descriptors. We find that the deep features from the convolutional feature maps are distinct from the traditional descriptors: the weighted sum of different level of features shows no superior performances than a simple summation of them. Kaiming et al. (2014) devise an approach called SPP (spatial pyramid pooling). In SPP, feature maps of the last convolutional layer are divided into a 3 or 4 scale pyramid. First the regional features in each scale are concatenated, then the scale-level features are concatenated to a fixed length vector to be forwarded to the next fully-connected layers. We find that this strategy is ineffective for unsupervised instance retrieval, leading to inferior performances compared to other simple combination methods (see the part about multi-scale representation in section 5.2 for more details.).

**Image representation using off-the-shelf CNNs**. Gong et al. (2014) propose the MOP (multi-scale orderless pooling) method to represent an image in which VLAD is used to encode the level 2 and level 3 features. Then features from different scales are PCA-compressed and concatenated to form the image features. This method is rather complicated and time-consuming. At the same time, Babenko et al. (2014) use Alexnet (Krizhevsky et al., 2012) trained on the Imagenet 1000-class classification task and retrain the network on task-related dataset. The retraining procedure gives a boost to the retrieval performances. Instead of using the output of the fully-connected layers as the image feature representations, Babenko & Lempitsky (2015) use the output feature maps of last convolutional layer to compute the image features. Recently, instead of sum-pooling the convolutional features, Tolias et al. (2015) use max-pooling to aggregate the deep descriptors. Their multi-scale method, called R-MAC (regional maximum activation of convolutions), further improves the previous results on four common instance retrieval datasets. Our work differs from these papers in that we explicitly explore the various factors that underpin the success of unsupervised instance retrieval, which have not been fully explored and analysed. By carefully choosing the different setting for each factor and combining them in a complementary way, we show that a large improvement can be achieved without additional cost.

## 3    IMPACTING FACTORS

When we employ off-the-shelf CNNs for the task of instance-level image retrieval, a natural question is: what kind of design choices should we make in order to make full use of the representational power of existing models? In this section, we summarize the five factors that may greatly impact the performance of the final image retrieval system. In section 5.2, we will show our experimental results on each key factor. Before we delve into the impacting factors, first we will give a brief introduction about how to represent an image using the activation feature maps of a certain layer.

### 3.1    CNN FEATURES FOR INSTANCE RETRIEVAL

In this paper, we are mainly interested in extracting compact and discriminative image features using the off-the-shelf CNNs in an efficient way. For a given image $I$, we simply subtract the mean value of the RGB channels from the original image and do not do other sophisticated preprocessing. Then the image is fed into the convolutional network and goes through a series of convolutions, non-linear activations and pooling operations. The feature activation maps of a certain layer can be interpreted as the raw image features, based on which we build the final image features. These feature maps form a tensor of size $K \times H \times W$, where $K$ is the number of feature channels, and $H$ and $W$ are height and width of a feature map. Each feature map represents a specific pattern which encodes a small part of information about the original image. If we represent the set of feature maps as $F = \{F_i\}, i = 1, 2, \ldots, K$, where $F_i$ is the $i^{th}$ activation feature map, then the most simple image feature is formulated as:

$$f = [f_1, f_2, \ldots, f_i, \ldots, f_K]^T. \tag{1}$$

In the above equation 1, $f_i$ is obtained by applying the feature aggregation method (see section 3.2) over the $i^{th}$ feature map $F_i$. Throughout this paper, we use feature maps after the non-linear activations (ReLU) so that the elements in each feature map are all non-negative. We also experiment with feature maps prior to ReLU, but find that they lead to inferior performances. After the image feature representation is obtained, post-processing techniques such as PCA and whitening can be further applied.

### 3.2    IMPACTING FACTORS ON PERFORMANCE

**Feature aggregation and normalization.**    After the feature maps of a certain layer are obtained, it is still challenging to aggregate the 3-dimensional feature maps to get compact vector representations for images. Previous papers use either sum-pooling (Babenko & Lempitsky, 2015) or max-pooling (Tolias et al., 2015) followed by $l_2$-normalization. Sum-pooling over a particular feature map $F_i$ is expressed as

$$f_i = \sum_{m=1}^{H} \sum_{n=1}^{W} F_i(m, n), i \in \{1, 2, \ldots, K\}, \tag{2}$$

while max-pooling is given by

$$f_i = \max_{m,n} F_i(m, n), \tag{3}$$

where $m, n$ are all the possible values over the spatial coordinate of size $H \times W$. In this paper, for the first time, different combinations of aggregation and normalization methods ($l_2$ and $l_1$ in the manner of RootSIFT (Arandjelović & Zisserman, 2012)) are evaluated and their results are reported.

**Output layer selection.**    Zeiler & Fergus (2014) has shown that image features aggregated from the feature activation maps of certain layers have interpretable semantic meanings. Gong et al. (2014) and Babenko et al. (2014) use the output of the first fully-connected layer to obtain the image features, while Babenko & Lempitsky (2015) and Tolias et al. (2015) use the output feature maps of the last convolutional layer. But these choices are somewhat subjective. In this paper, we extract dataset image features from the output feature maps of different layers and compare their retrieval performances. Based on the finding in this experiment, we choose the best-performing layer and also come up with a layer ensemble approach which outperforms state-of-the-art methods (see section 5.3).

**Image resizing.**    Famous models such as Alexnet (Krizhevsky et al., 2012) and VGGnet (Simonyan & Zisserman, 2014) all require that the input images have fixed size. In order to meet this requirement, previous papers (Gong et al., 2014; Babenko & Lempitsky, 2015) usually resize the input

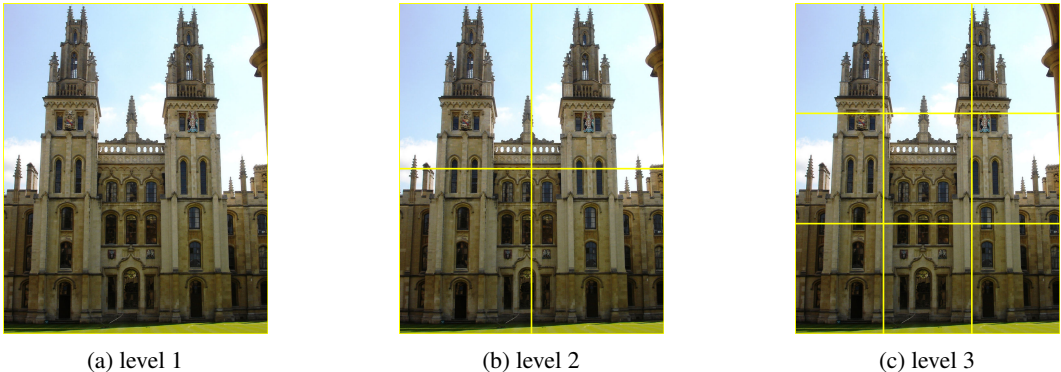

| (a) level 1 | (b) level 2 | (c) level 3 |

Figure 1: **An illustration of multi-scale representation of an image.** The whole image is divided into 3 levels from the coarsest (level 1) to the finest (level 3). At each level, the image is divided into different number of equal-sized regions.

images to the fixed size. We postulate that the resizing operation may lead to the distortion of important information about the objects in the natural images. Ultimately, this kind of operation may hurt the discriminative power of image features extracted from the network, thus degrading the retrieval performances. For the task of image retrieval, we think it is best to keep the images their original sizes and feed them directly to the network whenever possible. In this paper, three image resizing strategies are explored:

- Both the height and width of the dataset images are set to the same fixed value (denoted as *two-fixed*).

- The minimum of each dataset image's size is set to a fixed value. (The aspect ratio of the original image is kept.) (denoted as *one-fixed*).

- The images are kept their original sizes. (denoted as *free*).

**Multi-scale feature representation.** Unlike local feature descriptors such as SIFT (Lowe, 2004), the feature vector extracted from the deep convolutional networks for an image is a global descriptor which encodes the holistic information. When used for image retrieval, this kind of features still lack the detailed and local information desired to accurately match two images. Inspired by spatial pyramid matching (Lazebnik et al., 2006) and SPP (Kaiming et al., 2014), we explore the feasibility of applying this powerful method to obtain discriminative image features. An image is represented by a $L$-level pyramid, and at each level, the image is divided evenly into several overlapping or non-overlapping regions. The vector representations of these small regions are computed, then the regional vectors are combined to form the image feature vectors. The single scale representation of an image is just a special case of the multi-scale method in which the number of level $L$ equals 1.

Figure 1 shows an example of 3 level representations of an image. The time cost of re-feeding those small regions into the network to compute the regional vectors would be huge, thus unacceptable for instance retrieval tasks. Inspired by the work of Girshick (2015) and Tolias et al. (2015), we assume a linear projection between the original image regions and the regions in the feature maps of a certain layer. Then the regional feature vectors can be efficiently computed without re-feeding the corresponding image regions. In section 5.2, various settings for the multi-scale and scale-level feature combination methods are explored and their retrieval performances are reported and analysed.

**PCA and whitening.** Principal Component Analysis (PCA) is a simple yet efficient method for reducing the dimensionality of feature vectors and decorrelating the feature elements. Previous work (Babenko et al., 2014; Jégou et al., 2010) has shown evidences that PCA and whitened features can actually boost the performances of image retrieval. In this paper, we further investigate the usefulness of PCA and whitening within our pipeline and give some recommendations.

## 4 IMPLEMENTATION

We use the open source deep learning framework Caffe (Jia et al., 2014) for our whole experiments. The aim of this research is to investigate the most effective ways to exploit the feature activations of existing deep convolutional models. Based on past practices for networks to go deeper (Krizhevsky et al., 2012; Simonyan & Zisserman, 2014; Szegedy et al., 2015; He et al., 2015), a consideration for moderate computational cost, and also the results from Tolias et al. (2015) that deeper networks work better than shallower ones, we decide to use the popular VGG-19 model (Simonyan & Zisserman, 2014) trained on ImageNet as our model.

**Network transformation**. The original VGG-19 network only accepts an image of fixed size ($224 \times 224$), which is not the optimal choice when extracting image features for retrieval tasks. In order for the network to be able to process an image of arbitrary size (of course, the image size can not exceed the GPU's memory limit) and for us to experiment with different input image resizing strategies, we adapt the original VGG-19 network and change the fully-connected layers to convolutional (Long et al., 2015) layers. For more details about network transformations, see appendix A.

## 5 EXPERIMENTS

In this section, we first introduce the datasets used and the evaluation metrics. Then we report our experimental results for different impacting factors and give detailed analysis. In the last part, we show the performance of our method considering all these impacting factors and compare our method with the state-of-the-art methods on four datasets.

### 5.1 DATASETS AND EVALUATION METRICS

The **Oxford5k** dataset (Philbin et al., 2007) contains 5062 images crawled from Flickr by using 11 Oxford landmarks as queries. A total of 11 groups of queries — each having 5 queries with their ground truth relevant image list, are provided. For each query, a bounding box annotation is also provided to denote the query region. During experiment, we report results using the full query images (denoted as full-query) and image regions within the bounding boxes of the query images (denoted as cropped-query). The performance on this dataset is measured by mAP (mean average precision) over all queries.

The **Paris6k** dataset (Philbin et al., 2008) includes 6412 images[1] from Flickr which contains 11 landmark buildings and the general scenes from Paris. Similar to the Oxford5k dataset, a total of 55 queries belonging to 11 groups and the ground truth bounding boxes for each query are provided . The performance is reported as mAP over 55 queries.

The **Oxford105k**[2] dataset contains the original Oxford5k dataset and additional 100,000 images (Philbin et al., 2007) from Flickr. The 100,000 images are disjoint with the Oxford5k dataset and are used as distractors to test the retrieval performance when the dataset scales to larger size. We use the same evaluation protocol as the Oxford5k on this dataset.

The **UKB** dataset (Nistér & Stewénius, 2006) consists of 10200 photographs of 2550 objects, each object having exactly 4 images. The pictures of these objects are all taken indoor with large variation in orientation, scale, lighting and shooting angles. During experiment, each image is used to query the whole dataset. The performance is measured by the average number of same-object images in the top-4 results.

### 5.2 RESULTS AND DISCUSSION

In this section, we report the results of experiments on the impact of different factors and analyse their particular impact. The experiments in this section are conducted on the Oxford5k dataset.

**Feature aggregation and normalization.** In this experiment, we compare the different combinations of feature aggregation (sum-pooling and max-pooling) and normalization methods ($l_2$ and $l_1$)

---

[1] Following conventions, 20 corrupted images from this dataset are removed, leaving 6392 valid images.

[2] The image named "portrait_000801.jpg" was corrupted and manually removed from this dataset.

Table 1: **Comparison between different combinations of feature aggregation and normalization methods.**

| Method | full-query | cropped-query |
|--------|-----------|---------------|
| $max\text{-}l_1$ | 52.4 | 48.0 |
| $sum\text{-}l_2$ | 58.0 | 52.6 |
| $sum\text{-}l_1$ | 60.3 | 56.3 |
| $max\text{-}l_2$ | 60.1 | 53.5 |

Table 2: **Comparison between different image resizing strategies.** The numbers in the parentheses denote the sizes in which the maximum mAPs are achieved.

| Method | full-query | cropped-query |
|--------|-----------|---------------|
| *two-fixed* | 55.5 (864) | 38.7 (896) |
| *one-fixed* | 59.0 (800) | 39.3 (737) |
| *free* | 58.0 | 52.6 |

in terms of their retrieval performances. We use features from the layer conv5_4 with the *free* input image size. The results (%) are shown in Table 1. Sum-pooling followed by $l_1$ normalization leads to slightly better results than the other combinations, especially for the cropped-query. However, after preliminary experiment with a multi-scale version of $sum\text{-}l_1$ and $max\text{-}l_2$, we find that $max\text{-}l_2$ is much better than $sum\text{-}l_1$. For example, employing a 4 level representation of images in the Oxford5k dataset, for the case of full-query, we find that the mAP for the $max\text{-}l_2$ method is 65.1, while the mAP for $sum\text{-}l_1$ is only 51.3 (even lower than the single scale representation). Base on these results, we stick to $max\text{-}l_2$ in computing the final image features.

**Output layer selection.** In order to verify their feasibility for instance retrieval, we extract from the network the output feature maps of different layers and aggregate them to get the image feature vectors. We evaluate the performances using features from layer conv3_3 up to the highest fc7-conv layer (except the pooling layers, *i.e.* pool3, pool4 and pool5). Single-scale representations of the dataset images are used in this experiment.

Figure 2 shows the retrieval performances of image features corresponding to different layers. The retrieval performances for both the full and cropped queries increase as the layer increases from lower layer conv3_3 to higher layers and plateau in layer conv5_4 and fc6-conv, then the performances begin to decrease as the layers increase to fc7-conv. The result shows that features from lower layers such as conv3_3 and conv3_4 are too generic and lack the semantic meanings of the object in the image, thus rendering them unsuitable for instance retrieval. On the other hand, features from the highest layer (fc7-conv) contain the semantic meaning of objects but lack the detailed and local information needed to match two similar images. The best results are obtained in layer conv5_4 (0.601) and fc6-conv (0.618), where the feature vectors combine both the low-level detailed information and high level semantic meanings of the image. Based on these observations and the requirement for keeping the image features compact, we mainly focus on image features from the layer conv5_4 (dimensionality = 512 compared to 4096 of layer fc6-conv).

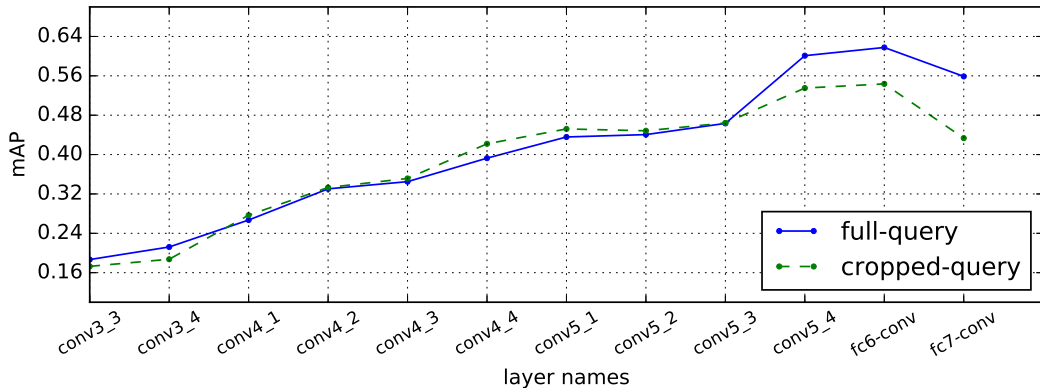

Figure 2: **Performance comparison between different layers.** This experiment is conducted using the *free* input image size.

**Image resizing.** We experiment with 3 kinds of image resizing strategies which are detailed in section 3.2. We use grid search to find the optimal size for the *two-fixed* and *one-fixed* strategy. As is shown in Table 2, the *free* input strategy outperforms or is close to the other two strategies: it

performs especially well in the cropped-query case. This experiment shows that changing the image aspect ratio (*two-fixed*) distorts the image information, thus reducing the performance dramatically. The *one-fixed* way is better than the *two-fixed* method. But information loss still occurs due to the resizing operation. The *free* method is able to capture more natural and un-distorted information from the images, which explains its superior performance over the other two methods. It is best to keep the images their original sizes for the instance retrieval tasks.

**The benefit of multi-scale representation.** In our multi-scale approach, the regional vectors from each scale are simply added together and $l_2$-normalized to form the scale-level feature vectors. Then features from different scales are combined and $l_2$-normalized to form the image representations. In fact, we also experimented with two methods which concatenate features from different scales. The first method is in same vein to spatial pyramid pooling (Kaiming et al., 2014), *i.e.*, region-level as well as the scale-level features are all concatenated to form a high dimensional vector. In the second method, region-level features are added while scale-level features are concatenated. We find that these two methods all lead to inferior results. The performance drop for the first in the case of cropped-query can be as large as 41%. The high dimensionality of the concatenated features (larger than 1.5k) will also lead to longer running times. Considering all these, we do not use concatenation of features in the following experiments.

Table 3: **Multi-scale representation: comparison between different methods.** "overlap" denotes whether the regions in each level (see Figure 1) have some overlapping areas. "s2","s3" mean that overlap occurs in level 2 or 3. "weighing" means if the features from each level are added using same weight or different weight. "version" means the different choice of the number of regions in each scale.

|      | scale | overlap | weighing | version | full-query | cropped-query |
|------|-------|---------|----------|---------|------------|---------------|
| (a1) | 2     | ×       | ×        | -       | 63.5       | 59.0          |
| (a2) | 2     | ×       | ✓        | -       | 63.9       | 61.0          |
| (b1) | 3     | ×       | ×        | -       | 64.2       | 60.9          |
| (b2) | 3     | ×       | ✓        | -       | 62.6       | 61.0          |
| (b3) | 3     | s2      | ×        | -       | 64.8       | 60.8          |
| (c1) | 4     | s3      | ×        | v1      | 65.1       | 61.4          |
| (c2) | 4     | s3      | ✓        | v1      | 64.8       | 60.7          |
| (c3) | 4     | s2,s3   | ×        | v1      | 65.5       | 60.8          |
| (c4) | 4     | s2,s3   | ×        | v2      | 65.9       | 61.5          |
| (c5) | 4     | s2,s3   | ✓        | v2      | 65.4       | 61.2          |
| (c6) | 4     | ×       | ×        | v3      | 64.5       | 61.3          |
| (c7) | 4     | s3      | ×        | v3      | 65.8       | 62.2          |
| (c8) | 4     | s2,s3   | ×        | v3      | **66.3**   | **62.6**      |

We conduct extensive experiments to decide the best configurations for the multi-scale approach and report our results in Table 3. First, we explore the impact of the number of scales on the retrieval performances. For the 2 and 3 scale representations, The region number for each level are $\{1 \times 1, 2 \times 2\}$, $\{1 \times 1, 2 \times 2, 3 \times 3\}$. For the 4 scale representation, 3 versions are used and they differ in the number of regions in each scale: for "v1", "v2", and "v3", the number of regions are $\{1 \times 1, 2 \times 2, 3 \times 3, 4 \times 4\}$, $\{1 \times 1, 2 \times 2, 3 \times 3, 5 \times 5\}$ and $\{1 \times 1, 2 \times 2, 3 \times 3, 6 \times 6\}$. Table 3 (a1)(b1)(c6) show the performances of using 2, 3, and 4 scales to represent the dataset images, respectively. Clearly, more scale levels improve the results and in the case of cropped-query, increase the performance by an absolute 2%.

We also conduct experiments to find whether the weighing of different scales leads to improved performance. The weighing method for features from different scales is similar to the manner of spatial pyramid matching (Lazebnik et al., 2006) — features from coarser level are given less weight while features from the finer levels are given more weight. Suppose the features of different scales for an $L$ scale representation are $f^1, f^2, \ldots, f^L$, then the image representation $f$ is expressed as:

$$f = \frac{1}{2^{L-1}} f^1 + \sum_{i=2}^{L} \frac{1}{2^{L-i+1}} f^i. \tag{4}$$

More details can be found in Lazebnik et al. (2006). Comparing the results of row (a1) and (a2), it seems that weighing different scales leads to better performance. But after more experiments, we find that the weighing method generally leads to inferior results as the number of scales increase,

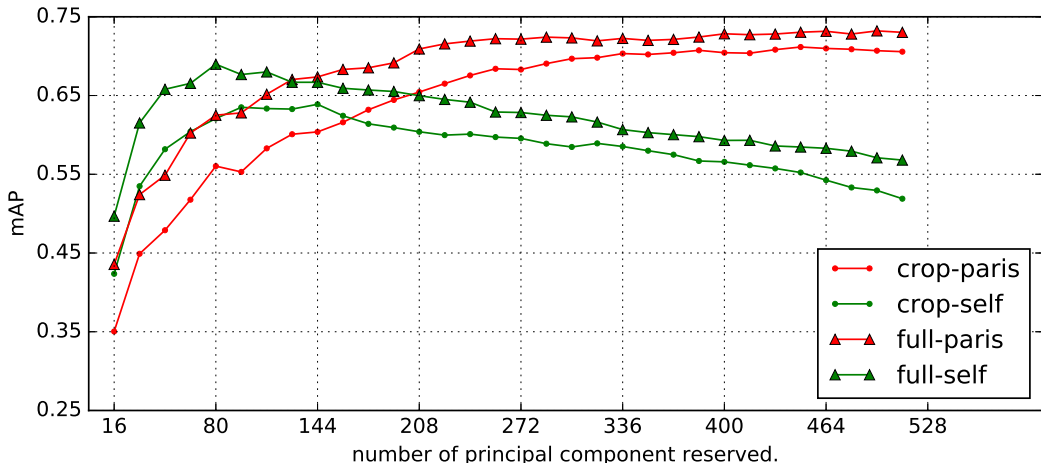

Figure 3: **The number of principal component reserved VS mAP.** We show the results of full and cropped query using the PCA and whitening matrix learned from the Oxford5k itself and Paris6k, denoted as "full-self", "full-paris" and "crop-self", "crop-paris".

*e.g.*, compare the results of row pair(b1)(b2) and (c1)(c2). These results suggest that deep features are different from the traditional local feature descriptors such as SIFT. We should exercise with caution when we apply the traditional wisdom found in SIFT to the deep convolutional descriptors, which is also suggested in Babenko & Lempitsky (2015). Based on the results of this experiment, no weighing methods are used in computing our final image feature representations.

Next, we look into the issue of overlapping between different scales and try to verify its usefulness. For each scale and its different versions, we set some overlapping areas between the neighboring regions in either one or two scales of the pyramid (For the exact configurations of overlap in all cases in Table 3, see appendix B for the complete descriptions). From the row pair (b1)(b3) and (c1)(c3), we can see that overlap increase the performance for full-query but decrease a little the performance for cropped-query. But for 4 scale v3 (note the pair(c7)(c8)), we see a consistent improvement for both the full and cropped queries. So we decided to use overlap in level 2 and 3 in computing our final features.

Table 4: **The impact of PCA and whitening.** "PCA on self" and "PCA on Paris" mean that the corresponding features are post-processed by the PCA and whitening matrices learned on the Oxford5k and Paris6k datasets, respectively. The numbers in the parentheses indicate the dimensionality of features used for obtaining the corresponding results.

| Feature | full-query | cropped-query |
|---|---|---|
| 3scale_overlap, original | 64.8 | 60.8 |
| 3scale_overlap, PCA on self | 65.4(80) | 60.9(112) |
| 3scale_overlap, PCA on Paris | 70.6(464) | 67.3(480) |
| 4scale_v3_overlap(s3), original | 65.1 | 61.4 |
| 4scale_v3_overlap(s3), PCA on self | 66.9(80) | 61.9(96) |
| 4scale_v3_overlap(s3), PCA on Paris | 72.3(464) | 70.8(496) |
| 4scale_v3_overlap(s2,s3),original | 66.3 | 62.8 |
| 4scale_v3_overlap(s2,s3), PCA on self | 69.0(80) | 63.9(144) |
| 4scale_v3_overlap(s2,s3), PCA on Paris | 73.2(496) | 71.2(448) |

**PCA and whitening**. We perform PCA and whitening for the features extracted from the Oxford5k dataset using the PCA and whitening matrix learned from the Oxford5k or the Paris6k dataset and $l_2$-normalize these features to get the final image representations.

The retrieval results for 3 groups of features (from Table 3(b3)(c1)(c8)) are shown in Table 4. Clearly, PCA and whitening lead to better performances. For all 3 groups of features, PCA and

Table 5: **Comparison with state-of-the-art methods.** "single" means multi-scale features from single layer (conv5_4) are used. "single, compression" uses the same features but compresses them to get the best performances. "layer ensemble" combines the similarity score from layer conv5_4 and fc6-conv. The dimensionality of the combined feature is set to 1024 for compactness considerations. All our methods use PCA and whitening.

| method | D | Oxford5k | | Paris6k | | Oxford105k | | UKB |
|---|---|---|---|---|---|---|---|---|
| | | full | cropped | full | cropped | full | cropped | |
| Jégou & Zisserman (2014) | 128 | - | 43.3 | - | - | - | 35.3 | 3.40 |
| Arandjelović & Zisserman (2012) | 128 | - | 44.8 | - | - | - | 37.4 | - |
| Jégou & Zisserman (2014) | 1024 | - | 56.0 | - | - | - | 50.2 | 3.51 |
| Razavian et al. (2014b) | 256 | 53.3 | - | 67.0 | - | 48.9 | - | 3.38 |
| Babenko et al. (2014) | 512 | 55.7 | - | - | - | 52.2 | - | 3.56 |
| Babenko & Lempitsky (2015) | 256 | 58.9 | 53.1 | - | - | 57.8 | 50.1 | 3.65 |
| Arandjelović et al. (2016) | 256 | 62.5 | 63.5 | 72.0 | 73.5 | - | - | - |
| Tolias et al. (2015) | 512 | - | 66.8 | - | 83.0 | - | 61.6 | - |
| ours (single) | 512 | 73.0 | 70.6 | 82.0 | 83.3 | 68.9 | 65.3 | 3.75 |
| ours (single, compression) | - | 73.2 | 71.2 | 83.0 | 84.0 | 68.9 | 65.8 | 3.76 |
| ours (layer ensemble) | 1024 | **75.6** | **73.7** | **85.7** | **85.9** | **71.6** | **69.2** | **3.81** |

whitening on the same dataset lead to insignificant improvement both in the case of full and cropped query. But after doing PCA and whitening on the Paris6k dataset, the results for both the full and cropped queries improve greatly. In fact, the improvement for the case of cropped-query is even more surprising. For example, for the third feature group, the improvement are 10.4% and 13.4% for the full and cropped queries. It should also be noted that as the the number of principal component reserved increases, the performance for "PCA on self" and "PCA on Paris" differs greatly. As is shown in Figure 3, the performance for the former peaks at a relatively low dimension (around 100) and begins to decrease, while for the latter, the performance increases as the number of principal component gets larger and then plateaus.

Do the above results mean that we should always compute the PCA and whitening matrix from any datasets other than the query dataset itself? The short answer is **no**. We find that for UKB, learning the PCA and whitening matrix on the Oxford5k dataset shows inferior results compared to learning the PCA and whitening matrix on UKB itself (about 2% drop in accuracy). This may be due to the large differences between the images of the two datasets as the Oxford5k dataset are mainly images of buildings while the images in UKB are mainly small indoor objects. We therefore recommend learning the PCA and whitening matrix on a similar dataset to achieve good performances.

## 5.3 Comparison with Other Methods

Based on the previous experimental results and our analysis of different impacting factors on the retrieval performances, we propose a new multi-scale image feature representation. For a given image in the dataset, the whole process of image feature representation is divided into two steps. First, the input image is fed into the network without the resizing operation (the *free* way) and a 4-scale feature representation is built on top of the feature maps of layer conv5_4. During the multi-scale representation step, max-pooling of feature maps are used and regional vectors from the same scale are added together and $l_2$-normalized. After that, features from different scales are summed and $l_2$-normalized again. The second step involves applying the PCA and whitening operations on features from the first step. The PCA and whitening matrix used are either learned from different or same dataset: specifically, for the Oxford5k and Oxford105k, it is learned in the Paris6k, while for Paris6k and UKB, it is learned on Oxford5k and UKB respectively. The final PCA and whitened image features are used for reporting our method's performances.

**Layer ensemble**. Inspired by previous work on model ensemble to boost the classification performances (Krizhevsky et al., 2012; Simonyan & Zisserman, 2014), we consider fusing the similarity score from different layers to improve the retrieval performances. Specifically, for two images, their similarity score is computed as the weighted sum of the scores from different layers (these weights sum to 1 so that overall similarity score between two images are still in the range $[0, 1]$.). We have evaluated various combination of layers to see their performances and find that best performance is achieved by combining the score from conv5_4 and fc6-conv. For the fc6-conv features of an image, we use a 3-scale representation as the size of output feature maps are already very small.

The fc6-conv features are compressed to low dimensional vectors for faster computation. Our layer ensemble achieves 75.6% and 73.7% on Oxford5k for the full and cropped queries respectively, showing a large improvement over previous methods. This suggests that features from the fc6-conv and conv5_4 are complementary. See Table 5 for the complete results on all four datasets.

**Comparison**. We compare the performance of our method with several state-of-the-art methods which use small footprint representations and do not employ the complicated post-processing techniques such as geometric re-ranking (Philbin et al., 2007) and query expansion (Arandjelović & Zisserman, 2012). The results are shown in Table 5. In all the datasets and different scenarios (full or cropped), our method achieves the best performance with comparable cost. For Oxford5k (cropped) and UKB dataset, the relative improvement of our best results over previous methods (from Tolias et al. (2015) and Babenko & Lempitsky (2015)) are 10.3% and 4.4%.

## 6 Conclusion

In this paper, we focus on instance retrieval based on features extracted from CNNs. we have conducted extensive experiments to evaluate the impact of five factors on the performances of image retrieval and analysed their particular impacts. Based on the insights gained from these experiments, we have proposed a new multi-scale image representation which shows superior performances over previous methods on four datasets. When combined with the technique "layer ensemble", our method can achieve further improvements. Overall, we have provided a viable and efficient solution to apply CNNs in an unsupervised way to datasets with a relatively small number of images.

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

## APPENDIX A   THE NETWORK TRANSFORMATIONS

In order for the network to process images of varying sizes, We change the layer fc6, fc7 and fc8 from the original model to fc6-conv, fc7-conv and fc8-conv. It should be noted there are certain constraints on the input image size due to the network's inherent design. The original network accepts an image of fixed size ($224 \times 224$), so the output feature maps of the last convolutional layer conv5_4 is of size $512 \times 7 \times 7$. As a result, when we change the operation between layer conv5_4 and fc6 from inner product to convolution, each filter bank kernel between conv5_4 and fc6-conv has size $7 \times 7$. This in turn means that if we are to extract features from layer fc6-conv and above, the minimum size of an input image must equal to or be greater than 224. For output feature maps of layer conv5_4 and below, there are no restrictions on the input image size. During the experiment, when we are extracting features from layer fc6-conv and above, the minimum size of an image is set to be 224 if it is less than 224.

## APPENDIX B   THE DETAIL OF OVERLAP IN EACH SCALE

In this paper, the overlaps between different regions occur in the 3 and 4 scale pyramid. A single region in each scale can be specified as the combination of a slice from the the width and height of the feature map. If a scale has $N \times N$ regions, then the number of slices in width and height of the feature map are both $N$. We use the same set of slices for both the width and height in this experiment.

In 3 scale (see Table 3 (b3)), overlap occurs only in scale 2, and the slice (in the proportion to the length of feature map width or height: $\{(0, \frac{2}{3}), (\frac{1}{3}, 1)\}$. In 4 scale v1 (Table 3 (c1)–(c3)), the slices for scale 2 and 3 are $\{(0, \frac{3}{4}), (\frac{1}{4}, 1)\}$ and $\{(0, \frac{2}{4}), (\frac{1}{4}, \frac{3}{4}), (\frac{2}{4}, 1)\}$. In 4 scale v2 (Table 3 (c4)(c5)), the slices for scale 2 and 3 are $\{(0, \frac{3}{5}), (\frac{2}{5}, 1)\}$ and $\{(0, \frac{3}{5}), (\frac{1}{5}, \frac{4}{5}), (\frac{2}{5}, 1)\}$. In 4 scale v3 (Table 3 (c6)–(c8)), the slices are $\{(0, \frac{4}{6}), (\frac{2}{6}, 1)\}$ and $\{(0, \frac{3}{6}), (\frac{1}{6}, \frac{4}{6}), (\frac{3}{6}, 1)\}$, for scale 2 and 3, respectively.

