# Peer review of "What Is the Best Practice for CNNs Applied to Visual Instance Retrieval?"

_ICLR 2017 — rejected_

[Reviewer Comment · AnonReviewer2 · 29 Nov 2016]
**Various, chunk 2**

5)
I agree with other reviewers on the lack of comparison with Gordo et al and Radenovic et al, though I understand that authors' arguments are that they do not want to train their networks (though then comparing with Arandjelovic et al. also doesn't make sense). It's still worth citing these papers and commenting on them. Also, with such an extensive set of experiments, it's a bit arguable if authors don't really do training - they don't do the canonical SGD, but they essentially perform grid search for parameters on the test (see question 3).

6)
I'm not sure what did we actually learn from this paper. To use the last conv? We knew that before as all recent papers do this (Arandjelovic et al, Tolias et al, Gordo et al, Radenovic et al, Babenko and Lempitsky, ..). That using original image sizes is important? We knew this as well, early works (Babenko et al 2014, etc) used smaller images while all recent works apply the networks convolutionally over original size images (e.g. Tolias et al have this experiment in table 1). That one should use PCA with whitening (and if possible learn whitening on the test set)? We knew this already as well. So the only two things that haven't been done in exactly the same way as people did it before is the multi-scale pooling (though obviously various other similar versions exist), and the exploration of max/sum pooling with l1 or l2 normalization (though the experiments in table 1 are basically ignored as sum-l1 works the best there, but authors then say that actually later they notice that for multiscale max-l2 works best). Actually the most interesting part for me, one that I can actually say I didn't know and don't think anyone knew, is figure 3.

7)
I think it's a bit of an overstatement to call this paper 'best practice for CNNs' when only a single CNN architecture, VGG-19, is considered. What is the best practice for other models, e.g. ResNet, Inception? Presumably the last conv is likely to be best though for ResNet it's not that clear, and I'm not sure if sum vs max pooling would change as those two networks were trained with sum pooling, and I'm not sure if any of the other conclusions hold either. This is more of a surgery of VGG-19 than best practices for CNNs in general.

8)
On a more philosophical level, and not only aimed at authors but also at others who are potentially reading this - this conference is about learning representations, while no learning is being performed. Taking CNNs as black boxes and tweaking the inputs and outputs in different ways with different normalizations is much more like using hand-engineered features like SIFT (replace black-box SIFT extractor with black-box CNN) than actually doing Deep Learning. I'm not saying this type of paper shouldn't exist as it's good to know what works best, but my preference in terms of what papers I would like to see in the future is:
a) There have been too many papers for using CNNs as black-boxes, I hoped we are finally over with this
b) For ICLR I think one should actually do some training, e.g. after we figure out the best image representation, now train the whole system end-to-end and see if you can improve the performance.
c) Design architectures which are specifically aimed at image retrieval - maybe something different than CNNs for classification pops up?
d) Figure out ways to train CNNs for retrieval, we know how to do it for classification by paying people to label millions of images, can we do something better for retrieval? (though this is to some extend addressed now by Arandjelovic et al, Gordo et al and Radenovic et al).


Other minor comments:

- I was also surprised by the "harder than category retrieval" statement, as reviewer 3. I wouldn't go as far as saying that the opposite is true either, the two just cannot be compared so easily.
- Inconsistencies of references (e.g. "Y. Lecun" vs "Ross Girshick", "CVPR" versus "Computer
Vision and Pattern Recognition", ..

[Official Review · AnonReviewer1 · rating 3 · confidence 5 · 15 Dec 2016]
**An outdated method with misleading claims.**

This paper explores different strategies for instance-level image retrieval with deep CNNs. The approach consists of extracting features from a network pre-trained for image classification (e.g. VGG), and post-process them for image retrieval. In other words, the network is off-the-shelf and solely acts as a feature extractor. The post-processing strategies are borrowed from traditional retrieval pipelines relying on hand-crafted features (e.g. SIFT + Fisher Vectors), denoted by the authors as "traditional wisdom".

Specifically, the authors examine where to extract features in the network (i.e. features are neurons activations of a convolution layer), which type of feature aggregation and normalization performs best, whether resizing images helps, whether combining multiple scales helps, and so on. 

While this type of experimental study is reasonable and well motivated, it suffers from a huge problem. Namely it "ignores" 2 major recent works that are in direct contradictions with many claims of the paper ([a] "End-to-end Learning of Deep Visual Representations for Image Retrieval" by  Gordo et al. and [b] "CNN Image Retrieval Learns from BoW: Unsupervised Fine-Tuning with Hard Examples" by Radenović et al., both ECCV'16 papers). These works have shown that training for retrieval can be achieved with a siamese architectures and have demonstrated outstanding performance. As a result, many claims and findings of the paper are either outdated, questionable or just wrong.

Here are some of the misleading claims: 

  - "Features aggregated from these feature maps have been exploited for image retrieval tasks and achieved state-of-the-art performances in recent years."
  Until [a] (not cited), the state-of-the-art was still largely dominated by sparse invariant features based methods (see last Table in [a]).
  
  - "the proposed method [...] outperforms the state-of-the-art methods on four typical datasets"
  That is not true, for the same reasons than above, and also because the state-of-the-art is now dominated by [a] and [b].
  
  - "Also in situations where a large numbers of training samples are not available, instance retrieval using unsupervised method is still preferable and may be the only option.".
  This is a questionable opinion. The method exposed in "End-to-end Learning of Deep Visual Representations for Image Retrieval" by Gordo et al. outperforms the state-of-the-art on the UKB dataset (3.84 without QE or DBA) whereas it was trained for landmarks retrieval and not objects, i.e. in a different retrieval context. This demonstrates that in spite of insufficient training data, training is still possible and beneficial.

  - Finally, most findings are not even new or surprising (e.g. aggregate several regions in a multi-scale manner was already achieved by Tolias at al, etc.). So the interest of the paper is limited overall.

In addition, there are some problems in the experiments. For instance, the tuning experiments are only conducted on the Oxford dataset and using a single network (VGG-19), whereas it is not clear whether these conditions are well representative of all datasets and all networks (it is well known that the Oxford dataset behaves very differently than the Holidays dataset, for instance). In addition, tuning is performed very aggressively, making it look like the authors are tuning on the test set (e.g. see Table 3). 

To conclude, the paper is one year too late with respect to recent developments in the state of the art.

[Official Review · AnonReviewer3 · rating 6 · confidence 4 · 16 Dec 2016]
**A paper with some good but limited and possibly slightly outdated experiments on object retrieval with CNNs**

The paper conducts a detailed evaluation of different CNN architectures applied to image retrieval. The authors focus on testing various architectural choices, but do not propose or compare to end-to-end learning frameworks.

Technically, the contribution is clear, particularly with the promised clarifications on how multiple scales are handled in the representation. However, I am still not entirely clear whether there would be a difference in the multi-scale settting for full and cropped queries.

While the paper focuses on comparing different baseline architectures for CNN-based image retrieval, several recent papers have proposed to learn end-to-end representations specific for this task, with very good result (see for instance the recent work by Gordo et al. "End-to-end Learning of Deep Visual Representations for Image Retrieval"). The authors clarify that their work is orthogonal to papers such as Gordo et al. as they assess instead the performance of networks pre-trained from image classification. In fact, they also indicate that image retrieval is more difficult than image classification -- this is because it is performed by using features originally trained for classification. I can partially accept this argument. However, given the results in recent papers, it is clear than end-to-end training is far superior in practice and it is not clear the analysis developed by the authors in this work would transfer or be useful for that case as well.

[Official Review · AnonReviewer2 · rating 3 · confidence 5 · 16 Dec 2016 (modified: 20 Jan 2017)]
**Not much utility in the paper**

Authors investigate how to use pretrained CNNs for retrieval and perform an extensive evaluation of the influence of various parameters. For detailed comments on everything see the questions I posted earlier. The summary is here:

I don't think we learn much from this paper: we already knew that we should use the last conv layer, we knew we should use PCA with whitening, we knew we should use original size images (authors say Tolias didn't do this as they resized the images, but they did this exactly for the same reason as authors didn't evaluate on Holidays - the images are too big. So they basically used "as large as possible" image sizes, which is what this paper effectively suggests as well), etc. This paper essentially concatenates methods that people have already used, and performs some more parameter tweaking to achieve the state-of-the-art (while the tweaking is actually performed on the test set of some of the tests).

The setting of the state-of-the-art results is quite misleading as it doesn't really come from the good choice of parameters, but mainly due to the usage of the deeper VGG-19 network. 

Furthermore, I don't think it's sufficient to just try one network and claim these are the best practices for using CNNs for instance retrieval - what about ResNet, what about Inception, I don't know how to apply any of these conclusions for those networks, and would these conclusions even hold for them. Furthermore the parameter tweaking was done on Oxford, I really can't tell what conclusions would we get if we tuned on UKB for example. So a more appropriate paper title would be "What are the best parameter values for VGG-19 on Oxford/Paris benchmarks?" - I don't think this is sufficiently novel nor interesting for the community.

[Final Decision · Program Chairs · 06 Feb 2017]
**ICLR committee final decision**

The paper conducts a detailed evaluation of different CNN architectures applied to visual instance retrieval. The authors consider various deep neural network architectures, with a focus on architectures pre-trained for image classification. 
 
 An important concern of the reviewers is the relevance of the evaluation given the recent impressive experimental results of deep neural networks trained end-to-end for visual instance retrieval by Gordo et al. "End-to-end Learning of Deep Visual Representations for Image Retrieval". Another concern is the novelty of the proposed evaluation given the evaluation of the performance for visual instance retrieval of deep neural network pre-trained for image classification performed in Paulin et al. "Convolutional Patch Representations for Image Retrieval: An Unsupervised Approach". 
 
 A revision of the paper, following the reviewers' suggestions, will generate a stronger submission to a future venue.